# Electroencephalogram-Based Subject Matching Learning (ESML): A Deep Learning Framework on Electroencephalogram-Based Biometrics and Task Identification

**DOI:** 10.3390/bs13090765

**Published:** 2023-09-14

**Authors:** Jin Xu, Erqiang Zhou, Zhen Qin, Ting Bi, Zhiguang Qin

**Affiliations:** 1School of Information and Software Engineering, University of Electronic Science and Technology of China, Chengdu 610097, China; 201611220101@std.uestc.edu.cn (J.X.); qinzhen@uestc.edu.cn (Z.Q.); qinzg@uestc.edu.cn (Z.Q.); 2Department of Computer Science, Maynooth University, W23 F2K8 Maynooth, Ireland

**Keywords:** EEG analysis, identify authentication, behavior recognition, deep learning

## Abstract

An EEG signal (Electroencephalogram) is a bioelectric phenomenon reflecting human brain activities. In this paper, we propose a novel deep learning framework ESML (EEG-based Subject Matching Learning) using raw EEG signals to learn latent representations for EEG-based user identification and tack classification. ESML consists of two parts: one is the $ESML_{1}$ model via an LSTM-based method for EEG-user linking, and one is the $ESML_{2}$ model via a CNN-based method for EEG-task linking. The new model ESML is simple, but effective and efficient. It does not require any restrictions for EEG data collection on motions and thinking for users, and it does not need any EEG preprocessing operations, such as EEG denoising and feature extraction. The experiments were conducted on three public datasets and the results show that ESML performs the best and achieves significant performance improvement when compared to baseline methods (i.e., SVM, LDA, NN, DTS, Bayesian, AdaBoost and MLP). The $ESML_{1}$ model provided the best precision at 96% with 109 users and the $ESML_{2}$ model achieved 99% precision at 3-Class task classification. These experimental results provide direct evidence that EEG signals can be used for user identification and task classification.

## 1. Introduction

The BCI (Brain–Computer Interface) system can convert the subject’s EEG signals into control commands or instructions for external devices. EEG signals are an efficient means to acquire brain signals corresponding to various electrical activities on the scalp surface area. There are some research works that have been conducted in the EEG field. Jirayucharoensak et al. use a deep learning network to detect emotion from nonstationary EEG signals, and they show that their method classifies three different levels of valence and arousal with an accuracy of 49.52% and 46.03%, respectively [1]. An et al. classify EEG data based on motor imagery tasks through applying a deep belief net (DBN); the recognition accuracy results were compared with Support Vector Machine (SVM), and the DBN classifier demonstrated better performance in all tested cases, with an improvement of 4–6% for certain cases [2]. Schirrmeister et al. use deep learning with convolutional neural networks (deep ConvNets) decoding and visualizing the informative EEG features; the method shows good performance as a widely used filter-bank common spatial patterns algorithm [3]. Thiago et al. propose a novel method for EEG representation based on deep learning; the results show that the method is a promising path to represent brain signals, overcoming baseline methods [4]. Mao et al. propose a new approach based on convolutional neural networks for EEG biometric identification; the approach achieved 97% accuracy from 100 subjects, and this work demonstrates the potential of deep learning solutions for real-life EEG-based biometric identification [5]. EEG also has many applications in BCI systems, such as medical rehabilitation [6,7,8,9,10], smart homes [11,12,13,14], education [15,16,17,18] and training [19,20,21,22,23], etc. In this work, a deep-learning-based framework called ESML (EEG-based Subject Matching Learning) is proposed for raw EEG signal processing to realize user identification and task classification. It does not need any EEG preprocessing operations, such as EEG denoising and feature extraction. The proposed framework is simple, but effective and efficient. ESML does not have any restrictions on thinking and motions for users when EEG is collected. Its robustness is tested using three EEG datasets, and it performs the best and achieves significant improvement when compared to baseline methods.

In the traditional identity authentication techniques, the access code, password and integrated circuit card are commonly used. However, they are vulnerable due to loss, forgery, theft or compromise since they are not bound to more secure human biological features. Biometric techniques as alternatives provide a more secure way for human identification, and they have been widely used in information systems and web application environments [24]. A biometric method being adopted for identity verification should be easy to distinguish and hard to imitate, specifically with the following desired properties [25]:Generality—Biometric data should be generalizable to every normal individual.Uniqueness—Users with different identities should be distinguishable via their unique biometrics.Stability—It should not change over time (long-term).Accessibility—It should be easily accessible, easily quantifiable and its acquisition should not be harmful to the individual.

The most common properties used in current identity recognition systems are mainly based on human biological characteristics, such as fingerprints, face recognition (both optical and infrared), iris scanning [26], DNA [27], keystroke entry patterns [28] and even gait [29]. However, they still have limited capability to deal with forgery. Some studies have shown that fake fingers made of gelatin can easily cheat fingerprint (FP) recognition systems. The false iris features of wax-engraved contact lenses can also make iris recognition systems hardly work. These data can be obtained from corpses, which are sometimes illegally used for identity verification. EEG-based identification systems can be promising and have outstanding performance. They are reliable and cost-effective biological data that are closely related to the human brain. On the one hand, EEG is a type of spontaneous electrical signal generated by the brain and recorded on the scalp of the subject. Since humans have unique brain structures, EEGs among subjects should be different—a high intersubject variability is expected. On the other hand, EEG is not only dependent on DNA, but also on life experience [30]. Compared with other biometric authentication technologies, EEG-based ones have the following advantages:Aliveness—EEG signals completely live with life and will disappear immediately if a subject dies.Stress-resistance (SR)—If a subject unwillingly accesses authentication systems under duress, this might incur a different pattern of EEG, which can potentially be detected.Anti-counterfeiting (AC)—Fingerprints can be found, especially when you leave them at many different systems. However, no one can obtain the brain signals of others.

The characteristics of different types of biometric techniques are summarized in Table 1.

Some studies have shown that EEG signals have unique patterns and are difficult to modify and replicate. Poulus et al., in using EEG signals for identity verification, used the neural network method to correspond particular EEGs to specific subjects [31]. Thorpe et al. proposed to use EEG signals collected from subjects under the status of resting to build a “thinking password” system for identity authentication [32]. They believed that when different users think about the same thing, their EEG signals are different. Salahiddin et al. used root mean square to generate EEG spatial patterns for identity recognition. They correctly identify up to 112 of the 122 subjects in their experiments [33]. Poulos et al. used an AR (Autoregressive) parameter to derive features and apply learning vector quantization in their neural networks model [34]. They obtained a classification accuracy of 72% to 84%. Isuru et al. used common spatial patterns as a method of feature and linear discriminant analysis to achieve a precision of 96.97% across 12 subjects [35]. Even though some algorithms achieve good results, there still exists a common problem: the identity authentication process is too complicated. Many previous EEG-based systems have five steps for EEG processing, including EEG acquisition, EEG denoising, feature extraction, model training and model validation. Figure 1 shows the general steps of EEG-based systems and the details of each step, as follows:1.EEG acquisition: It can be collected by electrodes placed on the scalp surface.2.EEG denoising: The noise in EEG signals during acquisition can be divided into eight categories: eye electrical (including blink signal), 50/60 power frequency interference, EEG, electrocardiogram, electrode loosening, sweating, breathing and pulse interference. Brain electrical signal denoising technology mainly includes the use of regression analysis, adaptive filter and direct phase subtraction, principal component analysis method, independent component analysis and wavelet transformation.3.Feature extraction: The most typical features used in EEG analysis are time and frequency, which can be obtained through many methods, such as power spectral density, wavelet transform and autoregressive model coefficients.4.Model training: Patterns can be learned through various classification models, such as support vector machines, nearest neighbors and naive Bayes.5.Model validation: The trained model is used for identity authentication and its performance is measured.

In this paper, we propose a deep learning-based framework called ESML without the need for data preprocessing operations, i.e., steps 2 and 3 in Figure 1. The raw EEG signals were used for model training. The contributions of this paper are summarized as follows:We introduce a deep learning-based framework called ESML, consisting of two neural networks. $ESML_{1}$ is an LSTM-based method used for EEG-based user identification, while $ESML_{2}$ is a CNN-based method used for EEG-based task classification.The proposed framework is simple, effective and efficient. ESML does not require any restrictions on EEG data collection and eliminates the need for EEG preprocessing operations.Experiments were conducted on three public EEG datasets, achieving an accuracy of up to 96% for the largest dataset with 109 users for EEG-user linking. Additionally, it achieved 99% precision in 3-Class task classification and 98% precision in the 5-Class case.

The rest of the paper is structured as follows: Section 2 introduces the background knowledge of brainwave signals. Section 3 provides a formal definition of EEG-based identity authentication and Section 4 presents the details of our framework, ESML. Section 5 shows the details of the EEG datasets and the baseline methods used in this paper. Section 6 discusses our experiments and corresponding results, followed by Section 7 summarizing the paper and outlining directions for future work.

## 2. Related Work

The genetic traits of human EEG have received great attention since the very beginning of human EEG recordings by Hans Verger in 1924 [36]. The human brain is an important part of the central neural system, including the cerebrum, cerebellum and brain stem. The cerebrum is the most complex component with the largest brain volume and the highest growth level. Different cortical regions control different nerve centers and undertake different tasks. Thus, each region of the cerebral cortex has its function. Researchers have standardized the placement of electrodes for collecting and recording brain waves. Jasper et al. proposed the 10–20 electrode system in 1958, which defines the electrode names for different positions of the head [37]. A modification termed the 10–10 system was proposed with 64 channels in 1994 [38]. In this paper, three public EEG datasets are used for our experiment and they are RSVP [39], Sternberg Task [40] and BIC2000 [41]. Figure 2 shows the topographic of the three datasets. Figure 2a,b show the 3D images of the electrode positions of the RSVP and Sternberg Task, respectively. They are generated by the EEG Pack tool (EEGLAB. http://sccn.ucsd.edu/eeglab). Figure 2c shows the electrode positions for the dataset of BIC2000 [41] and it used the international standard of 10–10 scalp electrode placement [38].

EEG signals stimulated by cerebral activities usually fall into several frequency bands: Delta, Theta, Alpha, Beta and Gamma bands. Each band contains signals associated with particular brain activities [42,43,44,45]. Delta band (0.5–4 Hz) represents the deep sleep state. Theta band (4–7.5 Hz) corresponds to the unconscious state of mind. Alpha band (8–13 Hz) corresponds to the state of calm and relaxation. Beta band (14–30 Hz) is related to thinking and problem-solving and Gamma band (30–45 Hz) is related to some pathology. The effect of different frequency bands on the experimental results has not yet been studied. In this study, a filtering operation will be used for EEG denoising on the baselines to make the results more equal to the comparison methods (see Section 5.2.1 for a detailed discussion).

## 3. Problem Definition

In this section, the EEG-based user identification and task classification problem will be formalized after introducing some definitions. $S=\{s_{1},s_{2},\cdots ,s_{N}\}$ is used to denote *N* subjects (users) and each subject performs *M* tasks. $T_{n}=\left\{t_{1},t_{2},\cdots ,t_{M}\right\}$ is used to denote *M* collection tasks for a subject *n*, where $t_{m}\left(m\in \left[1,M\right]\right)$ is a *K*-dimensional time series and every dimension represents an EEG signal for electrode placement (as shown in Figure 2). *L* is used to represent the length of the time series, and every task $t_{m}$ is a matrix of K×L. $T_{n}=\left\{{\tilde{t}}_{mk}^{l}\right\}\left(k\in \left[1,K\right],m\in \left[1,M\right],l\in \left[1,L\right]\right)$ is a 3D tensor denoting EEG signals from the placement *k* of user *n* at time *l* for the task *m*. $A=\{a_{1},a_{2},\cdots ,a_{Q}\}$ is used to denote the different activities for each subject. The overview of ESML is demonstrated in Figure 3.

## 4. Proposed Framework

This section presents the ESML framework in detail. Section 4.1 shows how to segment the EEG signals of each task horizontally and vertically for EEG-user identification and EEG-task classification, respectively. Section 4.2 introduces the proposed framework and Section 4.3 discusses the optimization algorithm.

### 4.1. EEG Segmentation

Since the length of each task $t_{mn}$ is relatively long, to reduce the computational complexity and capture richer user information from EEG, EEG will be split to improve the efficiency of EEG-user linking. The raw EEG task $t_{mn}$ will be devided into *r* consecutive sub-sequences $t_{mn}^{1},t_{mn}^{2},\cdots ,t_{mn}^{r}$ and the length of the each sub-sequence is:(1)$$\begin{array}{c} l=L/r \end{array}$$

Therefore, every sub-sequence $t_{mn}^{r}$ is a matrix of k×l. The schematic diagram of EEG data segmentation is shown in Figure 3. For task classification, segmenting EEG horizontally will decrease performance because the task characteristics existing in EEG will be disrupted. Thus, each task EEG will be divided vertically into sub-segments, with each $\in R^{1\times L}$.

### 4.2. EEG Characterization

In this paper, ESML achieves two objectives: EEG-user identification and EEG-task classification. A set of unlinked EEG signals will be linked to their corresponding users who generate them and classify tasks under which EEG was stimulated. The model $EMSL_{1}$ is for EEG-user linking and the model $ESML_{2}$ is for EEG-task linking.

#### 4.2.1. EEG-User Linking

For the EEG-user linking, one variant of the well-known Recurrent Neural Network (RNN) model, Long Short-Term Memory (LSTM) [46], will be used to control the input and output of identity. For the sub-EEG segmentation $T_{mn}=\left\{t_{mn}^{1},t_{mn}^{2},\cdots ,t_{mn}^{r}\right\}$, let $h_{t-1}$, $h_{t}$ and ${\tilde{h}}_{t}$ denote the last, current and candidate embedding state, respectively. The first model, $ESML_{1}$, has a total of five similar LSTM layers, in which the learning rate is 0.001 and the forgetting rate is 1.0. The LSTM model used in $ESML_{1}$ is implemented as follows [47]:(2)$$\begin{array}{cc} I_{t} & =\sigma (W_{I}t_{mn}^{i}+U_{I}h_{t-1}+V_{I}c_{t-1}+b_{I})\\ F_{t} & =\sigma (W_{F}t_{mn}^{i}+U_{F}h_{t-1}+V_{F}c_{t-1}+b_{F})\\ O_{t} & =\sigma (W_{O}t_{mn}^{i}+U_{O}h_{t-1}+V_{O}c_{t}+b_{O}) \end{array}$$
where $I_{t}$, $F_{t}$, $O_{t}$, and $b_{*}$ are, respectively, the input gate, forget gate, output gate and bias vector. σ is a logistic sigmoid function. Matrices *W*, *U* and *V* are different gate parameters. $t_{mn}^{i}$ is a segmentation of the EEG signal $T_{mn}$. The memory cell $C_{t}$ is updated by partially replacing the existing memory unit with a new cell $C_{t}$ as
(3)$$\begin{array}{c} C_{t}=F_{t}C^{t-1}+I_{t}\tanh\left(W_{C}t_{mn}^{i}+U_{C}h_{t-1}+b_{C}\right) \end{array}$$

The subject match learning is then updated by
(4)$$\begin{array}{c} h_{t}=O_{t}⊙\tanh\left(C_{t}\right), \end{array}$$
where tanh(·) refers to the hyperbolic tangent function and ⊙ is the entry-wise product.

#### 4.2.2. EEG-Task Linking

For the EEG-task linking model, 1D Convolutional Neural Networks (CNNs) are used on horizontally segmented tasks that contain the complete task information. The basic component of CNNs in $ESML_{2}$ is as follows:Input layer: The processed EEG signal $t_{mn}$ is 1D completed signal data 1×L from one channel in 1 min.Convolution layer: The convolutional layer tries to analyze each patch of a neural network to obtain more abstract features. ReLu is chosen as activation in the CNN part because of its simplicity and efficiency. We also add dropout operation in the last two layers in CNNs to avoid overfitting.Batch-norm layer: It is set up before the input of each convolution layer.Max-pooling layer: This operation is used to select the maximum element from the region of the feature map covered by the filter.

The hyperparameters used in the $ESML_{2}$ convolutional part are shown in Table 2.

#### 4.2.3. Linking

To link EEG to its user and tasks, the EEG representation ${\tilde{t}}_{mn}^{r}$ learned by the ESML models is fed into the softmax function:(5)$$\begin{array}{cc} {\tilde{t}}_{mn} & =softmax(W_{mn}h_{mn}+b_{mn})\\ \; & =\frac{\exp\{{t_{mn}}^{T}\kappa_{mn}\}}{\sum_{i=1}^{M}\sum_{j=1}^{N}\exp\left\{{t_{mn}}^{T}\kappa_{ij}\right\}} \end{array}$$
where κ is the set of parameters to be learned.

### 4.3. Optimization

EEG is unstable and contains high noises, adaptive moment estimation (Adam) optimization algorithm will be used in ESML [48]. Given an EEG sequence $T_{mn}=t_{mn}^{1},t_{mn}^{2},\cdots ,t_{mn}^{r}$ for task *m* and subject *n*, the ESML model will be trained to maximize the log-likelihood concerning κ:(6)$$\begin{array}{c} s\left(t_{mn}^{i}\right)\mapsto \sum_{t_{mn}^{i}\in S}\log f\left(s|t_{mn}^{i},\kappa \right) \end{array}$$
where *s* and S are, respectively, the ground-truth user of EEG $T_{mn}$ and the training data. At each step, Adam will be used to estimate the parameter set κ. Finally, the objective is to minimize the following cost function:(7)$$\begin{array}{c} \Phi \left(t_{mn}^{i},{\tilde{t}}_{mn}^{i}\right)=-\sum_{m=1}^{M}\sum_{n=1}^{N}\sum_{i=1}^{r}s\log\left({\tilde{t}}_{mn}^{i}\right) \end{array}$$
where ${\tilde{t}}_{mn}^{i}$ is the predicted vector representation. Parameters used in this paper for Adam are α=0.001 (stepsize), $\beta_{1}=0.9$, $\beta_{2}=0.999$ (exponential decay rates for the moment estimates) and $\epsilon =10^{-8}$ (avoiding zero values during iterations).

## 5. Experimental Design

This section presents the details of the experimental design. Section 5.1 shows the description of the three EEG public datasets. Section 5.2 introduces the details of the baseline methods. EEG denoising and EEG feature extraction methods for these baseline methods are also presented. Section 5.3 introduces the evaluation metrics used in this paper.

### 5.1. Datasets

In this paper, experiments were conducted on three public EEG datasets: RSVP [39], Sternberg Task [40] and BCI2000 [41]. The different datasets have different purposes for their original experiments, and the details are as follows:RSVP: This dataset was originally collected to explore the neural basis of target detection in the human brain, which was collected using a BIOSEMI Active View 2 system with 256 electrodes mounted on a whole-head elastic electrode cap (E-Cap Inc., Winsen, Germany) with a custom near-uniform montage across the scalp, neck and bony parts of the upper face. Computer data acquisition was performed via USB using a customized acquisition driver at a 256 Hz sampling rate with 24-bit digitization.Sternberg Task: The purpose of the Sternberg Task was to investigate event-related EEG dynamics through a variation of the Sternberg task. The Sternberg Task data were collected from 71 channels (69 scalp and two periocular electrodes, all referred to as right mastoid) at a sampling rate of 250 Hz with an analog passband of 0.01 to 100 Hz (SA Instrumentation, San Diego, CA, USA). Input impedances were brought under 5 kΩ by careful scalp preparation.BCI2000: BCI2000 was created and contributed to PhysioNet by the developers of the BCI2000 instrumentation systems. Users performed different motor/imagery tasks while 64-channel EEGs were recorded using the BCI2000 (http://www.bci2000.org) system.

Table 3 shows the statistical details of these three datasets.

### 5.2. Baselines

A comparison is drawn between some machine learning algorithms and our proposed framework ESML for EEG-user linking and EEG-task linking. The details of the baseline methods used here are as follows:SVM: Bashar et al. [49] used SVM to recognize humans from test EEG signals and obtained a true positive rate of 94.44%. In SVM implementation [49,50,51], the linear kernel is used for solving the EEG-based human recognition problem due to its better performance than other kernels such as RBF kernel and Gaussian kernel in our experiments.ConvNets: Robin et al. [3] used deep learning with convolutional neural networks for EEG decoding and visualization; their study thus shows how to design and train ConvNets to decode task-related information from raw EEG without handcrafted features and highlights the potential of deep ConvNets combined with advanced visualization techniques for EEG-based brain mapping. Convolutional Neural Networks are designed to recognize visual patterns directly from pixel images with minimal preprocessing. In machine learning, a ConvNet is a class of deep, feed-forward artificial neural networks that has successfully been applied to analyzing visual imagery.LDA: Isuru et al. [35] used linear discriminant analysis as a classification algorithm for their given set of user data, and the maximum accuracy recorded was 96.67%. The LDA algorithm [35,52] is a generalization of Fisher’s linear discriminant, a method used in statistics, pattern recognition and machine learning to find a linear combination of features that characterizes or separates two or more classes of objects or events.NN: Nearest neighbor [53,54] is the optimization problem of finding the point in a given set that is closest (or most similar) to a given point. In a previous work, Lee et al. [54] used Nearest neighbor (NN) classifier to obtain time and frequency characteristics in the EEG signals and achieved an accuracy of up to 95% for a dataset with seven users.DTS: Aydemir et al. proposed a decision tree structure-based method that was applied to EEG classification and achieved 55.92%, 57.90% and 82.24% classification accuracy rates on the test data of three subjects [55]. The decision tree is a map of the possible outcomes of a series of related choices and is a type of supervised learning algorithm that is mostly used in classification problems. It works for both categorical and continuous input and output variables.Bayesian: Bayesian classification algorithm is a statistical classification method, which is a class of algorithms using probability and statistics knowledge classification. Yu et al. [56] demonstrated that the Bayesian method they proposed achieved a better overall performance than the computing algorithms for EEG classification.AdaBoost: Hu [57] used the AdaBoost algorithm to recognize EEG signals, which is an iterative algorithm. The core idea is to train different classifiers on the same training set, and then combine these weak classifiers to form a stronger final classifier.MLP: Multi-layer Perceptron [51,58] is a forward-structured artificial neural network that maps a set of input vectors to a set of output vectors. MLP can be used as a directed graph, composed of multiple node layers, each layer is fully connected to the next layer.

To have a fair comparison between baselines and ESML, the steps of EEG denoising and feature extraction are used in the baseline methods. The details of the EEG denoising and feature extraction will be introduced in the next two sections: Section 5.2.1 and Section 5.2.2.

#### 5.2.1. EEG Denoising

To denoise EEG signals, the zero-mean method was used to normalize the raw EEG signal:(8)$$x{\left[n\right]}^{*}=\frac{x[n]-\mu [n]}{\sigma [n]}$$
where x[n] is the raw signal. μ[n] is the average of each channel EEG signal. σ[n] denotes the standard deviation of each channel EEG signal. $x{\left[n\right]}^{*}$ is the new signal after normalization. This would be useful in reducing the intra-subject variance of the EEG signals. An EEG signal has five major waves: Delta, Theta, Alpha, Beta and Gamma waves [59,60,61]. The EEG signal mainly ranges from 0.5 to 45 Hz. To remove artifacts and obtain better frequency characteristics, raw EEG signals are processed by filters, especially window pass filters, for frequencies 0.5–45 Hz.

#### 5.2.2. EEG Feature Extraction

In this paper, the Autoregressive Moving Average (ARMA) method and Power Spectral Density (PSD) [62] method will be used for EEG feature extraction. ARMA model [63] is a linear time-invariant system with excitation signals as white noise, which is used to describe the generalized stationary stochastic process. The AR process can be regarded as a full infinite impulse response filter, which can be described by the following difference equation:(9)$$\begin{array}{c} x\left(n\right)=c+\sum_{i=1}^{p}a\left(i\right)x\left(n-i\right)+e\left(n\right) \end{array}$$
where x(n) is a discrete random process, which represents the EEG signal. *c* is a constant, *p* is the order of the AR model, $a_{1},a_{2},...,a_{p}$ is the model coefficient and e(n) is discrete white noise. The time series signal EEG x(n) can be uniquely identified by $a_{1},a_{2},...,a_{p}$ and each EEG signal can be uniquely determined by the AR model coefficients. The PSD method is characteristic of extracting the EEG signal from the frequency domain. Since the real power spectral density function of the brain waves cannot be obtained by the limited sample data, the power spectral density of a stationary random signal can only be estimated using a given set of sample data. The non-parametric estimation method based on the Fourier transform of the correlation function is called the classical power spectrum estimation method, such as the periodic method and the Welch method. In this paper, the Welch method is used [64].

### 5.3. Evaluation Metrics

In this paper, the evaluation metric precision, recall and $F_{1}$ are used to measure the performance of models. They are commonly used for classification tasks. Specifically, the averages of these metrics are defined as follows:(10)$$Precision=\frac{\#\mathrm{correctly}\mathrm{identified}\mathrm{subject}}{\#\mathrm{all}\mathrm{identified}\mathrm{subject}}$$
(11)$$Recall=\frac{\#\mathrm{correctly}\mathrm{identified}\mathrm{subject}}{\#\mathrm{all}\mathrm{correctly}\mathrm{subject}}$$
and $F_{1}$ is the harmonic mean of the precision and the recall:(12)$$F_{1}=\frac{2\times P^{*}\times R^{*}}{P^{*}+R^{*}}$$
where $P^{*}$ and $R^{*}$ are, respectively, the precision and the recall averaged across all users in ESML.

## 6. Empirical Results

In this section, the empirical results will be discussed. Section 6.1 shows the results of the $ESML_{1}$ model on the EEG-user linking. Section 6.2 demonstrates the results of the $ESML_{2}$ model on the EEG-task linking. Section 6.3 provides a further discussion of the experimental results of the ESML model and a comparison with other works.

### 6.1. EEG-User Linking

The first experiment shows the effects of the size of EEG on the $ESML_{1}$ model for EEG-user linking. The experiment was conducted on the three EEG datasets. Figure 4 shows the effect of different sizes of segments of EEG on the EEG-subject linking precision. We can find that the different size of EEG has a minor effect on the EEG-subject linking precision, varying within a certain range. For the RSVP dataset, its precision varies from 0.17 to 0.29. For the Sternberg Task dataset, its precision varies from 0.60 to 0.75. For the BCI2000 dataset, its precision varies from 0.81 to 0.96. We also find that When $\tilde{k}$ is equal to the number of EEG acquisition channels, all three datasets achieve good accuracy. For the RSVP dataset, $\tilde{k}$ is chosen as 256. For the Sternberg Task dataset, $\tilde{k}$ is chosen as 72. For the BCI2000 dataset, $\tilde{k}$ is chosen as 64. For the analysis of the sampling length $\tilde{l}$, The best performance for the three different datasets was achieved within a range of intervals. The optimal sampling range of $\tilde{l}$ is 150–250 for the RSVP dataset. For the dataset of the Sternberg Task, the optimal sampling range is 50–150. For the dataset of BCI2000, the optimal sampling range is 120–160. In the following experiments, the $\tilde{l}$ value will be 150 for all of the datasets.

The second experiment here is to investigate the effect of training size on performance. The training/testing rates of 4:10, 7:7, 10:4 and 13:1 are chosen for the BCI2000 dataset. The cross-validation method was used here for model validation. The experimental results are shown in Figure 5 and Figure 6. We find that the EEG-user linking performance increases with the increasing number of training sizes. The $ESML_{1}$ model has the best performance at different training sizes compared to other baseline methods. Similar results were obtained for the other two datasets (i.e., RSVP and Sternberg Task).

Furthermore, Table 4 summarizes the performance of EEG-user linking among $ESML_{1}$ and baseline methods on the three datasets. We can find that the results for RSVP and Sternberg Task are less performant than that of BCI2000.

There are two reasons for this: one is that the sampling time of the RSVP and the Sternberg Task is too small; another is that the number of tasks for each subject when collecting EEG in the three datasets is different. These led to differences in performance. However, the $ESML_{1}$ model exhibited better performance than the baselines in three datasets, and the best precision rate which $ESML_{1}$ was able to achieve was 96% for the BCI2000 dataset.

### 6.2. EEG-Task Linking

In this part, the $ESML_{2}$ model will be analyzed and the BCI2000 data will be used for EEG-task linking. BCI2000 has 109 subjects and 14 task sessions for each subject. In these tasks, there were six different activities. The details of the BCI2000 tasks are shown in Table 5.

Previous research work [65] has shown that the same motor cortex is still activated even under imagination. The BCI2000 dataset will be categorized into 5-Class and 3-Class. For the 5-Class: the first two activities are regarded as one static class, and the other four activities are classified as the other four classes. For the 3-Class: activities $a_{1}$ and $a_{2}$ as one class, activities $a_{3}$ and $a_{4}$ as the second class, and activities $a_{5}$ and $a_{6}$ as the last class. The details of the 5-Class and 3-Class are shown in Table 6 and Table 7.

Table 8 shows the results of $ESML_{2}$ and baseline methods for EEG-task linking in 3-Class and 5-Class. We can see that the $ESML_{2}$ model can achieve 99% precision at the 3-Class case. It is superior to other baseline methods and achieves at least a 16% improvement over the 83% achieved by the best baseline method, SVM. In the 5-Class case, the $ESML_{2}$ model is still the best method compared to other baselines and achieved 98% precision.

### 6.3. Further Discussion

In the previous analysis, we can see that the proposed framework, ESML, has fewer EEG processing steps and can provide better performance than baseline methods (i.e., SVM, LDA, NN, DTS, Bayesian, AdaBoost and MLP). For the EEG-user linking, we also compare some other research works, and the details are shown in Table 9. We can find that these works have good results on EEG-user linking, but all of these works need to preprocess the EEG data and perform the feature extraction. Although the proposed $ESML_{1}$ model is not the best performance method in the table, it has a simpler EEG operation without preprocessing (i.e., denoising and feature extraction). Moreover, the $ESML_{1}$ model achieved the best precision rate, at 96%, with 109 users, and the number of users is much higher than that of other research works. For the EEG-task linking, D. La Rocca et al. [66] tested their algorithm called Mahalanobis distance-based classifier and claimed a 100% accuracy on the same BCI2000 dataset. However, they built a binary classification model only for eyes-closed (i.e., $a_{1}$ in Table 5) and eyes-open (i.e., $a_{2}$ in Table 5) resting state conditions. The $ESML_{2}$ model proposed in this paper can achieve 99% precision for the 3-Class case and 98% precision for the 5-Class case. It can provide better performance compared to other baseline methods.

## 7. Conclusions

In this paper, a deep learning-based framework, ESML, was proposed for raw EEG signal processing to realize user identification and task classification. The proposed framework is simple but effective and efficient. It does not have any restrictions on thinking and motions for users during EEG collection and it does not require any EEG preprocessing operations, such as EEG denoising and feature extraction. For the ESML framework, it consists of two models. One is the $ESML_{1}$ model via the LSTM-based method for EEG-user linking. Another one is the $ESML_{2}$ model via CNN-based method for EEG-task linking. $ESML_{1}$ model can provide the best precision, at 96%, with 109 users, while $ESML_{2}$ model achieved a 99% precision for the 3-Class case and a 98% precision for the 5-Class case. The experiments provide direct evidence which indicates that EEG signals can be used for user identification and task classification. In the three public EEG datasets, ESML can perform the best and achieve significant improvement when compared to baseline methods. Although this study shows promising results for EEG-based user identification and task classification, developing more sophisticated models is always a worthwhile pursuit as a future direction. In future work, we would like to develop a real-time system that can enable us to observe EEG features, thus helping people to better understand brain activity.

## Figures and Tables

**Figure 1 behavsci-13-00765-f001:**
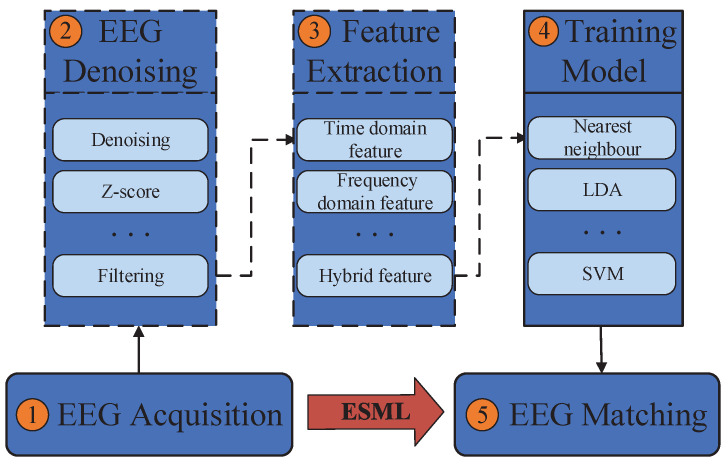
General steps of EEG-based systems.

**Figure 2 behavsci-13-00765-f002:**
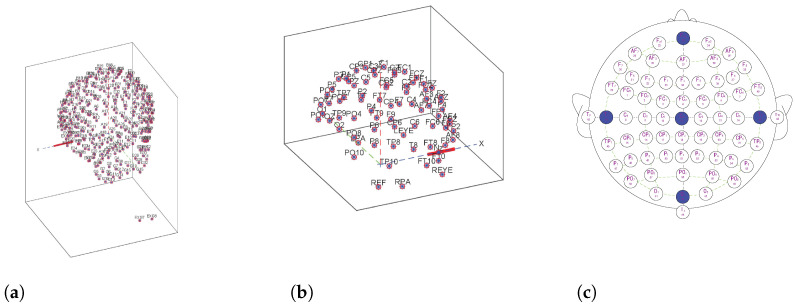
Topographic maps of the three public EEG datasets. (**a**) RSVP, (**b**) Sternberg Task, (**c**) BCI2000.

**Figure 3 behavsci-13-00765-f003:**
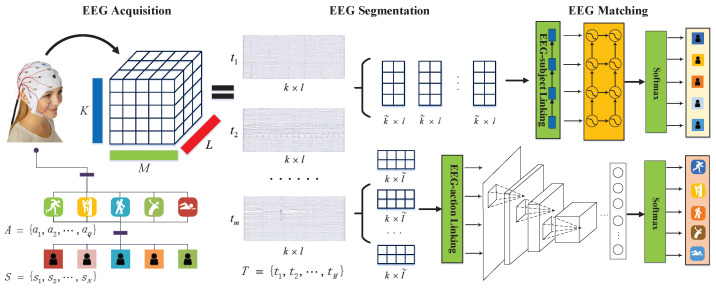
Overview of the ESML Framework. For EEG-user linking, we first acquire EEG signals for an individual on *m* tasks. Each task is a matrix $\in R^{k\times l}$, where *k* is the dimensionality of signal representation for each time point and *l* is the time interval. EEG signals for each task are further divided into *r* sub-segments $\in R^{k\times \tilde{l}}$.

**Figure 4 behavsci-13-00765-f004:**
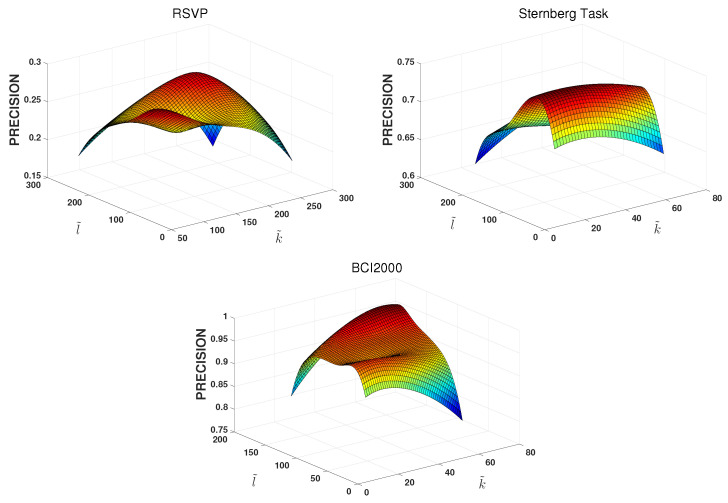
Precision comparison for different sizes of segmentation for EEG-user linking. $\tilde{l}$ represents the length of time series, $\tilde{k}$ denotes the number of EEG acquisition channel.

**Figure 5 behavsci-13-00765-f005:**
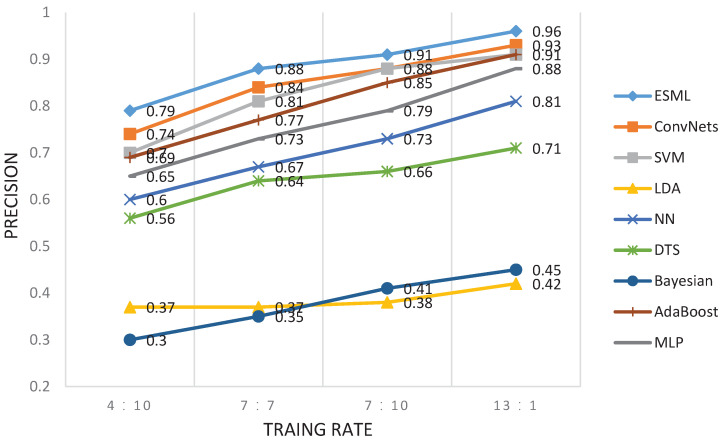
Precision comparison of overall EEG-user linking precision for different methods under different training/testing rates.

**Figure 6 behavsci-13-00765-f006:**
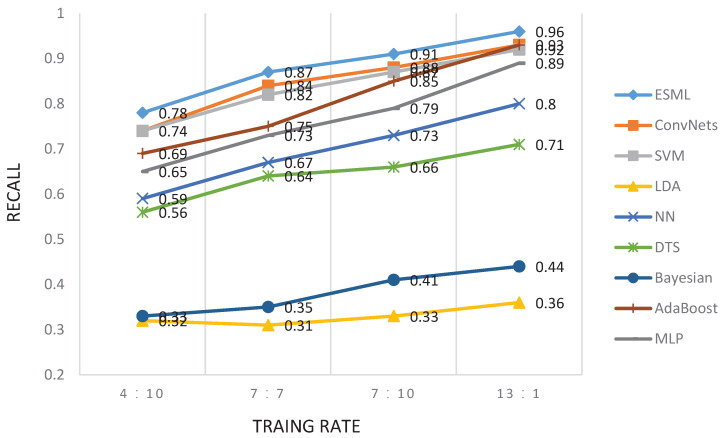
Recall comparison of overall EEG-user linking precision for different methods under different training/testing rates.

**Table 1 behavsci-13-00765-t001:** Comparison of characteristics for different types of biometric techniques. FP: fingerprint; SR: stress-resistance; AC: anti-counterfeiting.

Characteristics	EEG	FP	Face	Iris	Voice
Generality	*√*	*√*	*√*	*√*	*√*
Uniqueness	*√*	*√*	*√*	*√*	*√*
Stability	*√*	*√*	*√*	*√*	*√*
Accessibility	*√*	*√*	*√*	*√*	*√*
Aliveness	*√*	×	×	×	×
SR	*√*	×	×	×	×
AC	*√*	×	×	×	×

**Table 2 behavsci-13-00765-t002:** The hyperparameters setting in $ESML_{2}$

Layer	Convolution					Pooling		
	Filters	Kernel Size	Stride	Padding	Output Dim	Pool Size	Strides	Output Dim
1	16	3	1	Same	9600×1	[1,2]	[1,2]	4800×16
	16	3	1	Same	9600×16			
2	32	3	1	Same	4800×32	[1,2]	[1,2]	2400×32
3	64	3	1	Same	2400×64	[1,2]	[1,2]	1200×64
4	128	3	1	Same	1200×128	[1,2]	[1,2]	600×128
5	128	3	1	Same	600×128	[1,2]	[1,2]	300×128

**Table 3 behavsci-13-00765-t003:** Data description and statistics. *N*: the number of users; *M*: the number of tasks per user; *F*: the frequency of the EEG signal; *K*: the number of channels.

Dataset	*N*	*M*	*F*(Hz)	*K*
RSVP	7	2	256	256
Sternberg Task	23	4	256	72
BCI2000	109	14	160	64

**Table 4 behavsci-13-00765-t004:** Performance comparison of $ESML_{1}$ with baseline methods for EEG-user linking.

Methods	RSVP			Sternberg Task			BCI2000		
	Precision	Recall	*F*1	Precision	Recall	*F*1	Precision	Recall	*F*1
$ESML_{1}$	**0.37**	**0.30**	**0.29**	**0.75**	**0.74**	**0.71**	**0.96**	**0.96**	**0.96**
SVM	0.23	0.34	0.27	0.72	0.58	0.56	0.93	0.92	0.92
ConvNets	0.28	0.26	0.27	0.71	0.67	0.70	0.93	0.93	0.93
LDA	0.15	0.19	0.16	0.45	0.44	0.44	0.42	0.36	0.36
NN	0.28	0.31	0.29	0.67	0.66	0.64	0.81	0.80	0.80
DTS	0.30	0.33	0.30	0.61	0.59	0.57	0.71	0.71	0.70
Bayesian	0.16	0.14	0.15	0.42	0.42	0.40	0.45	0.44	0.43
AdaBoost	0.33	0.23	0.22	0.70	0.72	0.69	0.93	0.93	0.92
MLP	0.25	0.27	0.25	0.63	0.67	0.62	0.91	0.89	0.89

**Table 5 behavsci-13-00765-t005:** Activity description for BCI2000 dataset.

Activity ID	Activity Description	Task ID
$a_{1}$	Resting state with open eyes	$t_{1}$
$a_{2}$	Resting state with closed eyes	$t_{2}$
$a_{3}$	Open and close left or right fist	$t_{3},t_{7},t_{11}$
$a_{4}$	Imagine opening and closing left or right fist	$t_{4},t_{8},t_{12}$
$a_{5}$	Open an close both fists or both feet	$t_{5},t_{9},t_{1}3$
$a_{6}$	Imagine opening and closing both fists or both feet	$t_{6},t_{10},t_{14}$

**Table 6 behavsci-13-00765-t006:** Task classification for BCI2000 5-Class.

Class ID	1	2	3	4	5
Task ID	$t_{1},t_{2}$	$t_{3},t_{7},t_{11}$	$t_{4},t_{8},t_{12}$	$t_{5},t_{9},t_{1}3$	$t_{6},t_{10},t_{14}$

**Table 7 behavsci-13-00765-t007:** Task classification for BCI2000 3-Class.

Class ID	1	2	3
Task ID	$t_{1},t_{2}$	$t_{3},t_{7},t_{11},t_{4},t_{8},t_{12}$	$t_{5},t_{9},t_{1}3,t_{6},t_{10},t_{14}$

**Table 8 behavsci-13-00765-t008:** Performance comparison of $ESML_{2}$ with baseline methods for EEG-task linking.

Method	3-Class			5-Class		
	Precision	Recall	*F*1	Precision	Recall	*F*1
$ESML_{2}$	**0.99**	**0.99**	**0.99**	**0.98**	**0.98**	**0.98**
SVM	0.83	0.82	0.82	0.78	0.78	0.78
LDA	0.37	0.34	0.35	0.24	0.23	0.23
NN	0.83	0.83	0.83	0.78	0.78	0.78
DTS	0.63	0.63	0.63	0.51	0.51	0.51
Bayesian	0.44	0.42	0.26	0.16	0.21	0.20
AdaBoost	0.78	0.76	0.76	0.70	0.69	0.69
MLP	0.79	0.79	0.35	0.76	0.75	0.75

**Table 9 behavsci-13-00765-t009:** Comparison with other research works.

Research Work	EEG Feature	Method	Number of Users	Performance
Polus et al. [31]	FFT	LVQ	45	Correct score: 80% to 100%
Isuru et al. [67]	IHAR	KNN	12	Accuracy: 99.0±0.8%
Gui et al. [68]	WT	ANN	32	Correct score: 90%
Brigham et al. [69]	AR	SVM	6	Accuracy: 99.76%
Isuru et al. [35]	CSP	LDA	12	Accuracy: 96.97%
Proposed work	×	ESML	109	Precision: 96%

## Data Availability

Not applicable.

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
