# Peer review of "Electroencephalogram-Based Subject Matching Learning (ESML): A Deep Learning Framework on Electroencephalogram-Based Biometrics and Task Identification"

_behavsci, 2023, doi:10.3390/bs13090765_

Round 1

Reviewer 1 Report

This study aimed to propose an DL approach for task classification using EEG data. I have the following major suggestions.

What is the novelty of this study although several DL approaches for task and user classification have been proposed earlier?

The abstract should be improved by combining the objectives, short methodology, main findings, and prospective application.

Authors should include conceptual figures of their proposed DL approach with more details and model parametrization.

Captions should be more detailed.

Data preprocessing methods must be accompanied with references.

EEG is highly sensitive to the powerline, muscular, and cardiac artifacts. In EEG data preprocessing, authors need to mention how you handle AC power, ECG, eye-blinks, and EMG artifacts in EEG signals. Do the authors think that their proposed method is robust to such kinds of artifacts?

Authors should introduce the ML/DL applications in disease, and mental workload prediction in broad scope, such as article, big-ecg: cardiographic predictive cyber-physical system for stroke management; in article, explainable artificial intelligence model for stroke prediction using eeg signal; in article, healthsos: real-time health monitoring system for stroke prognostics; in article, quantitative evaluation of task-induced neurological outcome after stroke; in article, driving-induced neurological biomarkers in an advanced driver-assistance system; in article, quantitative evaluation of eeg-biomarkers for prediction of sleep stages.

Both testing ROC curves need to be shown for each task class. What model validation method authors used?

Model performance matrices should be reported for task and biometric classification separately.

I recommend using heatmap (deepSHAP/GradCAM) to explain the contribution of EEG features in the models.

The discussion section needs to be added. Authors must make discussion on the advantages and drawbacks of their proposed method with other studies adding a table in the discussion section.

EEG-domain explanation of heatmap findings needs to be described in support of reference in discussion.

From the writing point of view, the manuscript must be checked for typos and the grammatical issues should be improved.

NA

Reviewer 2 Report

1) Abstract needs to be rewritten and highlight the novelty of the work.

2) Add the contribution of your work at the end of the introduction section.

3) Proposed algorithm is missing. Add the proposed algorithm.

4) This work uses a number of equations but not mentioned the purpose and role of these equations.

5) How the model is trained and dataset description is missing. 

6) Why author print the graph of precision and recall? All the information shown in figure 5 and 6 is also available in table 4 and 5. Remove the figure 5 and 6 or justify the reason for including these figures.

7) compare the performance of your model with the existing model.

8) Add the finding in the conclusion section.

9) add the future work in the conclusion section.

4)

No

Round 2

Reviewer 1 Report

Author needs to address following major issues from last review comments:

The abstract should be improved by combining the objectives, short methodology, main findings, and prospective application.

 Both testing ROC curves need to be shown for each task class. What model validation method authors used?

 I recommend using heatmap (deepSHAP/GradCAM) to explain the contribution of EEG features in the models.

 The discussion section needs to be added. Authors must make discussion on the advantages and drawbacks of their proposed method with other studies adding a table in the discussion section.

 EEG-domain explanation of heatmap findings needs to be described in support of reference in discussion.

 From the writing point of view, the manuscript must be checked for typos and the grammatical issues should be improved.

Need major English revision

Reviewer 2 Report

The author has addressed all the comments. 

A minor proofread is required.

Round 3

Reviewer 1 Report

Most of review comments still pending to address. 

NA